Do N-arachidonyl-glycine (NA-glycine) and 2-arachidonoyl glycerol (2-AG) share mode of action and the binding site on the β2 subunit of GABAA receptors?

Baur Roland
Gertsch Jürg
Sigel Erwin erwin.sigel@ibmm.unibe.ch
Institute of Biochemistry and Molecular Medicine, University of Bern , Bern , Switzerland
Rocha Joao
Electronic publication date: 2013 Sep 10
Publication date: 2013
Volume: 1
Electronic Location ID: e149
Received 2013 Jun 25; Accepted 2013 Aug 14
Copyright: © 2013 Baur et al.
Copyright year: 2013
Copyright holder: Baur et al.
License: This is an open access article distributed under the terms of the Creative Commons Attribution License, which permits unrestricted use, distribution, and reproduction in any medium, provided the original author and source are credited.
License URL: https://creativecommons.org/licenses/by/3.0/

Keywords: GABAA receptors, Endocannabinoids, GABA, 2-AG, NA-glycine

Funding: Swiss National Science Foundation 31003A_132806/1 31003A_120672 This work was supported by the Swiss National Science Foundation grants 31003A_132806/1 (ES) and 31003A_120672 (JG). The funders had no role in study design, data collection and analysis, decision to publish, or preparation of the manuscript.

==============================
NA-glycine is an endogenous lipid molecule with analgesic properties, which is structurally similar to the endocannabinoids 2-AG and anandamide but does not interact with cannabinoid receptors. NA-glycine has been suggested to act at the G-protein coupled receptors GPR18 and GPR92. Recently, we have described that NA-glycine can also modulate recombinant α1β2γ2 GABAA receptors. Here we characterize in more detail this modulation and investigate the relationship of its binding site with that of the endocannabinoid 2-AG.

Introduction

GABA is the major inhibitory neurotransmitter in mammalian brain. Its fast effects are mediated by synaptic and extrasynaptic GABAA receptors. These receptors are composed of five subunits that surround a central chloride ion channel (Macdonald & Olsen, 1994; Sieghart, 1995; Sieghart & Sperk, 2002; Sigel & Steinmann, 2012). The major receptor isoform consists of α1, β2, and γ2 subunits (Olsen & Sieghart, 2008). Different approaches led to the widely accepted 2α:2β:1γ subunit stoichiometry (Chang et al., 1996; Tretter et al., 1997; Farrar et al., 1999; Baumann, Baur & Sigel, 2001; Baumann, Baur & Sigel, 2002; Baur, Minier & Sigel, 2006) with a subunit arrangement γβαβα anti-clockwise as seen from the synaptic cleft (Baumann, Baur & Sigel, 2001; Baumann, Baur & Sigel, 2002; Baur, Minier & Sigel, 2006). The pharmacological properties are dependent on subunit composition (Sigel et al., 1990) and arrangement (Minier & Sigel, 2004).

Neurosteroids (Belelli & Lambert, 2005) and the endocannabinoid 2-AG (Sigel et al., 2011) have been documented as endogenous ligands of GABAA receptors. A binding site for 2-AG has been shown to be present specifically on each of the two β2 subunits present in a pentameric receptor. Moreover it has been proposed that 2-AG dips into the membrane and binds to the fourth trans-membrane sequence (M4) of β2 subunits (Baur et al., 2013).

NA-glycine has been shown to be present in amounts of about 50 pmol/g dry weight in rat brain tissue and about 140 pmol/g dry weight in spinal cord (Huang et al., 2001). This can be compared to the levels of 2-AG that have been reported to be 4 and 50 nmol/g wet weight in brain (Sigel et al., 2011) and spinal cord (Guasti et al., 2009), respectively. Thus, NA-glycine is about 1000-fold less abundant than 2-AG. NA-glycine has no functional affinity for CB1 receptors (Sheskin et al., 1997), but may activate the G protein coupled receptors GPR18 (Kohno et al., 2006) and GPR92 (Oh et al., 2008) and target Na+/Ca2+-exchanger NCX (Bondarenko et al., 2013) and T-type Ca2+-channels (Barbara et al., 2009). Interestingly, NA-glycine exerts analgesic properties in different rodent models of pain (Huang et al., 2001; Succar, Mitchell & Vaughan, 2007). Recently, it has been reported that NA-glycine can also interact with glycine receptors, potentiating α1 and interestingly inhibiting α2 and α3-containing glycine receptors (Yevenes & Zeilhofer, 2011). It should be noted that NA-glycine functionally interacts with glycine receptors with an apparent affinity >10 µM. This should be compared to the functional affinities to GABAA receptors, NCX and T-type Ca2+-channels that are estimated <10 µM.

We have previously demonstrated that NA-glycine allosterically potentiates GABAA receptors (Baur et al., 2013), but it remained unclear whether this occurred via the same binding site as 2-AG. The aim of the present study was therefore to establish whether NA-glycine acts at the identical site on GABAA receptors as 2-AG. In most studied aspects Na-glycine acts similar to 2-AG, indicating a common binding site. However, some point mutations that abrogate modulation by 2-AG leave the initial modulation by NA-glycine nearly unaffected and only over time the modulation is gradually decreased to zero. Differential solubilization effects of 2-AG and NA-glycine may account for this phenomena. Thus, at least during initial phases of interaction with GABAA receptors the mode of binding is different for the two compounds.

Methods

Material

2-AG and NA-glycine were obtained from Cayman Chemical (Chemie Brunschwig, Basel, Switzerland). All other chemicals, unless mentioned otherwise below, were from Sigma (Buchs, Switzerland).

Expression of GABAA receptors in Xenopus oocytes

Capped cRNAs were synthesized (Ambion, Austin, TX, USA) from the linearized plasmids with a cytomegalovirus promotor (pCMV vectors) containing the different subunits, respectively. A poly-A tail of about 400 residues was added to each transcript using yeast poly-A polymerase (United States Biologicals, Cleveland, OH, USA). The concentration of the cRNA was quantified on a formaldehyde gel using Radiant Red stain (Bio-Rad Laboratories, Reinach, Switzerland) for visualization of the RNA. Known concentrations of RNA ladder (Invitrogen, Life Technologies, Zug, Switzerland) were loaded as standard on the same gel. cRNAs were precipitated in ethanol/isoamylalcohol 19 : 1, the dried pellet dissolved in water and stored at −80°C. cRNA mixtures were prepared from these stock solutions and stored at −80°C. Xenopus laevis oocytes were prepared, injected and defolliculated as described previously (Sigel, 1987; Sigel & Minier, 2005; Animal Permit No. BE98/12, Kantonaler Verterinärdienst, Kanton Bern). They were injected with 50 nL of the cRNA solution containing rat wild type α1 and wild type or mutated β2 and wild type γ2 subunits at a concentration of 10 nM:10 nM 50 nM (Boileau et al., 2002) and then incubated in modified Barth’s solution at + 18°C for at least 24 h before the measurements. Where indicated concatenated subunits α1-β2-α1/γ2-β2 or α1-β2-α1/γ2-β1 or α1-β1-α1/γ2-β2 or α1-β1-α1/γ2-β1 were used at a concentration of 25 nM:25 nM, each.

Functional characterization of the GABAA receptors

Currents were measured using a modified two-electrode voltage clamp amplifier Oocyte clamp OC-725 (Warner Instruments, Camden, CT, USA) in combination with a XY-recorder (90% response time 0.1 s) or digitized at 100 Hz using a PowerLab 2/20 (AD Instruments) using the computer programs Chart (ADInstruments GmbH, Spechbach, Germany). Tests with a model oocyte were performed to ensure linearity in the larger current range. The response was linear up to 15 µA.

Electrophysiological experiments were performed using the two-electrode voltage clamp method at a holding potential of −80 mV. The perfusion medium contained 90 mM NaCl, 1 mM KCl, 1 mM MgCl2, 1 mM CaCl2, 5 mM Na-HEPES (pH 7.4) and 0.5% DMSO and was applied by gravity flow 6 ml/min. The perfusion medium was applied through a glass capillary with an inner diameter of 1.35 mm, the mouth of which was placed about 0.4 mm from the surface of the oocyte. Allosteric modulation via the 2-AG site was measured at a GABA concentration eliciting about 1% of the maximal GABA current amplitude (EC1). In each experiment, 1 mM GABA was applied to determine the maximal current amplitude. Subsequently increasing concentrations of GABA were applied until 0.5%–1% of the maximal current amplitude was elicited (0.3–3 µM). For modulation experiments, GABA was applied for 20 s alone or in combination with 2-AG or NA-glycine. 2-AG or NA-glycine were pre-applied for 30 s. Modulation of GABA currents was expressed as (I(modulator+GABA)/IGABA–1) ∗ 100%. Inhibition by DEA was determined at the end of a 1 min co-application with either NA-glycine or 2-AG following a 30 s pre-application of both compounds. The perfusion system was cleaned between drug applications by washing with dimethylsulfoxide to avoid contamination.

Determination of critical micelle concentrations (CMC)

Assays were performed as reported previously (Raduner et al., 2007). In brief, compounds (from 2 mM stock solutions) were incubated at increasing concentrations with 0.1 nM fluorescein (free acid, 99%, Fluka, Switzerland) for 90 min at 30°C in Nanopure distilled water. Experiments were carried out on 96-well microtiter plates (excitation at 485 nm, emission at 535 nm) on a TECAN Farcyte reader. Experiments were performed in triplicates in three independent experiments and data are mean values ± S.D.

Results

Both NA-glycine and 2-AG allosterically potentiate recombinant α1β2γ2 GABAA receptors expressed in Xenopus oocytes. Both compounds share the arachidonoyl tail structure but differ in their hydrophilic head groups (Fig. 1). Please note that at physiological pH, NA-glycine is negatively charged. Here we wanted to compare the GABAA receptor binding site for NA-glycine with the well-characterized binding site for 2-AG.

Figure 1 Chemical structure of NA-glycine and 2-AG.

Figure 2A shows current traces of a cumulative concentration–response curve of the allosteric potentiation of α1β2γ2 GABAA receptors at a GABA concentration of 1 µM. At the highest concentration used the current trace displayed the typical signs of an open channel block, rapid apparent desensitization and an off current. This phenomenon made a precise curve fit impossible as maximal potentiation could not be determined precisely. The averaged concentration–response curve (Fig. 2B) was fitted with the assumption of different maximal potentiation. From these fits it was estimated that the EC50 was between 1 and 10 µM (not shown). Direct activation by 3 µM NA-glycine elicited no significant current (<2 nA) in oocytes where 100 µM GABA elicited a current >7 µA.

Figure 2 Concentration-dependent potentiation of currents mediated by recombinant α1β2γ2GABAA receptors.

(A) Receptors were expressed in Xenopus oocytes and currents were measured by using electrophysiological techniques at a GABA concentration eliciting 0.5–1.0% of the maximal current amplitude (EC0.5–1.0). GABA was applied twice (single bars) and subsequently in combination with increasing concentrations of NA-glycine. The numbers above the double bars indicate the concentration of NA-glycine in µM. NA-glycine was pre-applied for 30 s. Original current traces are shown. (B) shows the averaged concentration-dependent potentiation of currents elicited by GABA by NA-glycine. Four experiments as shown under (A) were averaged. Data are shown as mean ± SD (n = 4). Such an averaged curve has been shown before based on 3 experiments and missing the point at 10 µM NA-glycine (Baur et al., 2013).

Allosteric potentiation by 3 µM NA-glycine was determined at different concentrations of the endogenous agonist GABA. Figure 3 shows that the degree of potentiation was rapidly decreasing with increasing concentrations of GABA. The comparable properties of 2-AG are also shown in Fig. 3. We tried to rationalize these findings using a model that has previously been proposed on the basis of other observations (Baumann, Baur & Sigel, 2003; Fig. 4A). This model assumes binding of GABA to two sites differing in their binding affinity and transition to the open state with low propensity of singly ligated states and high propensity of the doubly ligated state. Figure 4B shows computed current amplitudes in dependence of the GABA concentration. In addition, a predicted curve is shown where it is assumed that NA-glycine promotes transition of the singly ligated receptor from the closed to the open state. Figure 4C shows a computed GABA concentration-dependence of the current potentiation expected in this case. The model predicts that sizeable potentiation is limited to very low concentrations of GABA.

Figure 3 Influence of the GABA concentration.

Current potentiation by 3 µM NA-glycine (closed circles) or 2-AG (closed squares) was determined at different concentrations of GABA. Potentiation decreased with increasing concentrations of GABA. The GABA concentration response curve was fitted with a mean EC50 of 35 µM and a mean Hill coefficient of 1.5 (not shown).

Figure 4 Mode of action of NA-glycine.

Simplified model (A) The model assumes two agonist binding sites 1 and 2 with different affinities. 2-AG affects the closed/open transition of the two singly ligated states. The receptor R can first bind GABA (A) either to the site 1 (AR) or the site 2 (RA). The receptor occupied by two agonist molecules ARA can isomerize to the open state ARA*, the receptors occupied by a single agonist molecule can isomerize to the open states AR* and RA*. Binding is described with K as dissociation constants and gating with L as closed state/open state equilibrium. (C) Theoretical GABA concentration response curves in the absence and presence of NA-glycine. The following parameters were assumed: 0.24 for L, 10 and 2.2 for L1 in the absence and presence of NA-glycine, respectively, 11 and 2.4 for L2 in the absence (line) and presence (dashed line) of NA-glycine, respectively, 30 µM for K1, 90 µM for K2. (C) Dependence of the potentiation by NA-glycine on the concentration of GABA obtained by the ratio of the computed current in the presence of NA-glycine divided by the current in its absence.

We have previously shown that the CB1 receptor ligand DEA antagonizes potentiation by 2-AG (Baur et al., 2013). Therefore, we compared the ability of DEA to antagonize potentiation by NA-glycine and 2-AG. Based on the structural similarity of the three compounds we assumed a competitive behaviour. Figure 5 compares the cumulative concentration inhibition curves for 3 µM NA-glycine and for 3 µM 2-AG. Potentiation by NA-glycine was inhibited half-maximally at 72 ± 36 µM (n = 4) DEA and potentiation by 2-AG at 1.4 ± 0.6 µM (n = 6). If the two ligands displayed a similar apparent affinity for potentiation at the same site where DEA acts, providing equal water solubility and lipid solubilization, a similar inhibitory potency of DEA would have been expected. In order to investigate if DEA and NA-glycine act competitively, we repeated a concentration inhibition curves at 6-times lower concentration (0.5 µM) of NA-glycine. Half-maximal inhibition was observed at 96 ± 41 µM (n = 4) DEA (Fig. 5). This could be interpreted as non-competitive interaction of DEA with NA-glycine. Similarly, we performed concentration inhibition curves at 1 and 15 µM 2-AG. The higher concentration of 2-AG elicits direct current that amount to less than 0.1% of the maximal current amplitude elicited by GABA in the same oocytes. As expected for a competitive interaction between 2-AG and DEA, half-maximal inhibition was shifted to the left upon decrease of the 2-AG concentration from 3 µM to 1 µM, but the inhibition curve became much flatter, indicating the DEA becomes partially inactive at higher concentrations. Half-maximal inhibition for 15 µM was not reached at concentrations up to 100 µM (n = 4) DEA (Fig. 5). In case of a non-competitive interaction of DEA with 2-AG an IC50 of about 1.4 µM and in case of a competitive interaction an IC50 of about 7 µM would be expected in the latter case. The observed results cannot be explained by classical receptor theory and we therefore speculate that 2-AG and NA-glycine exhibit a different water solubility and lipid solubilisation in the experimental setup. In the discussion we mention possible explanations.

Figure 5 Concentration inhibition curve of DEA.

Increasing concentrations of DEA were co-applied with 1 µM 2-AG (open circles), 3 µM 2-AG (open squares), 15 µM 2-AG (open diamonds), 0.5 µM NA-glycine (filled circles), or 3 µM NA-glycine (filled squares). Data are shown as mean ± SEM (n = 4).

NA-glycine shows higher efficacy than 2-AG for potentiation of currents elicited by GABA. 2-AG is metabolically stable in Xenopus oocytes as no degradation by serine hydrolases was found (not shown). In case NA-glycine competes for the same binding site as 2-AG and both molecules have a similar apparent affinity to this binding site, it would be anticipated that the degree of potentiation by both agents at the same concentration would result in an intermediate potentiation as compared to the individual agents. The apparent affinity of 2-AG has been determined as 2 µM, while the apparent affinity of NA-glycine is estimated 1–10 µM here. As shown in Fig. 6A, combined application results surprisingly in nearly the same extent of potentiation as application of NA-glycine alone.

Again, this may be caused by a differential water solubility and membrane solubilisation behaviour of NA-glycine and 2-AG. We therefore measured the critical micelle concentrations of both molecules. The apparent CMC was >100 µM for NA-glycine and 4.2 ± 0.5 µM for 2-AG, pointing to significant self-assembly and detergent behaviour of 2-AG.

As 2-AG fails to potentiate in GABAA receptors where the β2 subunit is replaced by β1, we tested potentiation by NA-glycine in α1β1γ2 receptors (Fig. 6B). Similarly to 2-AG, potentiation by NA-glycine depends on the presence of β2 subunits. We studied potentiation by NA-glycine in concatenated receptors containing either two β2 subunits, two β1 subunits or one each β1 and β2 in different positions in the receptor pentamer (Fig. 7). Receptors containing two β2 subunits exhibited strong potentiation while receptors containing two β1 subunits showed very weak potentiation. Intermediate potentiation was observed in receptors containing one each, β1 and β2. This strongly indicates that the NA-glycine binding site is located on the β2 subunit as previously shown with 2-AG (Sigel et al., 2011).

Figure 6 Effect of subunit combination and co-application with 2-AG.

(A) Current potentiation by the combined application of 3 µM NA-glycine and 3 µM 2-AG is compared with the individual application of the two substances. (B) Current potentiation by 3 µM NA-glycine in α1β2γ2 receptors and α1β1γ2 receptors. Potentiation is strongly dependent on the presence of the β2 subunit.

Figure 7 Concentration-dependent potentiation of currents mediated by concatenated GABAA receptors.

Concatenated α1-β1-α1/γ2-β1, α1-β1-α1/γ2-β2, α1-β2-α1/γ2-β1 or α1-β2-α1/γ2-β2 receptors were expressed in Xenopus oocytes and currents were measured at a GABA concentration eliciting 0.5–1.0% of the maximal current amplitude (EC0.5−−1.0). Current potentiation by increasing concentrations of NA-glycine was determined. Four such experiments were averaged. Data are shown as mean ± SD (n = 4).

A number of point mutations have been described to interfere with the potentiation by 2-AG. We tested the effect of the point mutations β2W428C, β2S429C, β2F432C, β2F439L and β2V443C. Original current traces are shown for the mutant receptor α1β2S429Cγ2. These traces are compared with traces from wild type receptors (Fig. 8A). While wild type receptors show a time-independent potentiation by NA-glycine, mutant receptors showed initially a potentiation that rapidly decayed over time. As these mutant receptors show a similar dependence on GABA as wild type receptors, and the experiment were carried out at very low GABA concentrations this current transient is not due to desensitization. In the case of 2-AG the effect of these mutations is a reduction of the potentiation independent of the time of exposure to 2-AG. This behaviour is observed with NA-glycine for the potentiation of α1β2F432Cγ2 receptors, but not the other mutant receptors studied.

Figure 8 Effect of point mutations that reduced potentiation by 2-AG on the potentiation of NA-glycine.

(A) Potentiation by 3 µM NA-glycine is compared between wild type receptors and receptors containing the point mutation S429C in the β2 subunit. This mutation results at the beginning of the drug application in an about 50% reduction of potentiation and after 1 min drug application potentiation is abolished. (B) Wild type receptors are compared with mutant receptors. Current potentiation is indicated at the beginning of the drug application (filled bars) and after 1 min drug exposure (open bars).

The mutation studies indicate a site of action in the inner leaflet of M4 of the β2 subunit. In this case NA-glycine has to traverse the lipid bilayer either by diffusion or mediated by a transport system and this may require some time. In order to test the time-dependence of action of NA-glycine we exposed an oocyte to GABA followed by GABA and NA-glycine (Fig. 9). Indeed, onset of modulation was slow and did not reach a steady level within 1 min. Upon switch of the medium to GABA only, a slow decay of the potentiation was observed.

Discussion

NA-glycine allosterically potentiates GABAA receptors like the major endocannabinoid 2-AG. We aimed at localizing the site of interaction of Na-glycine with recombinant α1β2γ2 GABAA receptors relative to the site for 2-AG. An interpretation of our results is hampered by the fact that the apparent affinity for the potentiation by NA-glycine could not be determined accurately. However, we can estimate this value to be in the range of 1–10 µM, which compares well with the value of 2 µM for 2-AG (Sigel et al., 2011). The fact that we find significant potentiation of GABAA receptors by >0.1 µM NA-glycine may reflect the better water solubility of NA-glycine over 2-AG at low concentrations and even indicate biological relevance as the average in vivo concentration in the central nervous system may be estimated from the dry tissue content as about 15–50 nM and NA-glycine is unlikely to be randomly distributed.

The following observations argue for a similar mode of action of NA-glycine and 2-AG. First, both substances only act exclusively at low GABA concentration (Fig. 3), putatively by enhancing the opening of singly ligated receptor channels (Fig. 4). A leftward shift of the concentration response curve for GABA as observed in the case of benzodiazepines does not abrogate potentiation below EC50 (Sigel & Steinmann, 2012). To our knowledge, this is a new mode of action of a ligand. Second, investigation of receptors with different β subunits (Fig. 5) and experiments with concatenated receptors containing either no, one, or two β2 subunits (Fig. 6) strongly indicate that both ligand binding sites are located on the β2 subunit. A common binding site in the inner leaflet of the fourth trans-membrane region (M4) of this subunit is suggested by the fact that modulation by both agents is either reduced or abolished in five identical mutant receptors, at least in the late phase of action of NA-glycine (Fig. 8B). The onset of action for both substances was found to be slow (Fig. 9; Baur et al., 2013). On the basis of these observations, it is tempting to assume a common binding site for the two ligands.

Figure 9 Time course of the potentiation by NA-glycine.

An oocyte expressing α1β2γ2 receptors was sequentially exposed to medium alone, to 1 µM GABA, to the same concentration of GABA in combination with 5 µM NA-glycine, to 1 µM GABA alone and the to medium. This experiment was repeated two more times with similar results.

The following observation cannot be explained by classical receptor theory in case NA-glycine and 2-AG use an identical binding site, i.e., display a similar apparent affinity and interact with each other in a competitive way. Combined application of two compounds with similar affinities at identical concentrations is then expected to result in an intermediate potentiation as compared to that by individual compounds. Instead, the observed potentiation is similar to the one by NA-glycine alone. A second observation is difficult to reconcile with a common binding site for NA-glycine, 2-AG and the inhibitor of the potentiation of 2-AG, DEA. Namely, DEA prevents potentiation by NA-glycine only at 50-fold higher concentrations as that caused by 2-AG. As mentioned in the result section, the interaction between DEA and 2-AG cannot be explained by classical receptor theory. Since NA-glycine exerts a significant higher CMC than 2-AG, differential solubilisation of NA-glycine and 2-AG with Xenopus oocytes may account for some of the effects observed in this study. The way these lipids are organized in an aqueous environment will affect entry of the molecules into the bilayer, binding equilibrium, and the way the receptor is occupied. If this holds true the observations with co-application of NA-glycine and 2-AG as well as the inhibition of NA-glycine and DEA have to be seen in a new light. In spite of our observations the three agents could still all bind to largely overlapping sites within an extended surface able to bind flexible hydrophobic structures.

The mutant receptors α1β2W428Cγ2, α1β2S429Cγ2, α1β2F439Lγ2 and α1β2V443Cγ2 all largely abrogate modulation by NA-glycine after 1 min of combined application of GABA with NA-glycine. This abrogation is not present at the beginning of the combined application, but sets in rather slowly. We have no explanation for this observation. Solubility considerations do not help to explain this phenomenon.

Overall, most arguments point to a similar action and possibly overlapping binding site for NA-glycine and 2-AG. No matter what the exact mode of interaction of NA-glycine with the GABAA receptor is, this agent represents by far the stronger positive allosteric modulator than 2-AG, although the latter is more abundant in brain. The implications of our findings for the analgesic effect of NA-glycine remain to be studied. Abbreviations

GABA γ-aminobutyric acid

GABAA receptor γ-aminobutyric acid type A receptor

NA-glycine N-arachidonyl-glycine

2-AG 2-arachidonoyl glycerol

DEA docosatetraenylethanolamide

We thank Dr. V Niggli for carefully reading the manuscript and Dr. A Chicca for determining stability of 2-AG in Xenopus oocytes.

Additional Information and Declarations

Competing Interests

Author Contributions

Animal Ethics

The authors declare no competing interests.

Roland Baur performed the experiments, analyzed the data.

Jürg Gertsch contributed reagents/materials/analysis tools, wrote the paper.

Erwin Sigel conceived and designed the experiments, analyzed the data, wrote the paper.

The following information was supplied relating to ethical approvals (i.e., approving body and any reference numbers):

Kantonaler Verterinärdienst, Kanton Bern

Approval No. BE98/12.

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
