# Peer review of "Do N-arachidonyl-glycine (NA-glycine) and 2-arachidonoyl glycerol (2-AG) share mode of action and the binding site on the β2 subunit of GABAA receptors?"

_PeerJ, doi:10.7717/peerj.149_

## Round 0.1 · original submission · Minor Revisions

Dear Dr Sigel ,
your manuscript has been revised by 4 experts in the field. Three reviews have raised only minor questions that you can found below. However, one of the reviewers has stated that “The present paper submitted to the PeerJ offers only 2 small additions to these previously published facts “, referring to previous paper from you lab. I realize that this is the most serious comment that you should give a good rebuttal to this reviewer. Furthermore, the same reviewer have asked about “Data shown on this plot are probably containing the same values as in J.Neurochem paper 2013 (Baur et al) in Fig 3 (n=3)” and “The course of the potentiation by 2-AG shown in J.Neurochem paper 2013 Fig.5 looks very similar (the legend is almost identical) to figure 9 of the present paper (this is shown now for the NA-glycine)”. These two points should be clarified and/or modified by you. The reviewer also stated that some parts of the introduction are identical to parts of your previous paper published in J. Neurochem., though this is not so critical as the two other points cited above, try to change the text a little. In addition to these critical aspects, the reviewer has also indicated that you should consider the following reference “Yevenes and Zeilhofer, PlosOne, 2011” and several other questions that you should consider to improve your manuscript. In the case you do not agree with the reviewer, please, give a clear rebuttal to the reviewer(s).

Reviewer 1 ·

Basic reporting

„Do N-arachidonyl-glycine (NA-glycine) and 2-arachidonoyl glycerol (2-AG) share mode of action and the binding site on the beta2 subunit of GABAA receptor?“ by Baur et al is a continuation of a row of papers from the same group, who discovered that the GABAA receptor can be potentiated by physiologically relevant concentrations of the endocannabinoid 2-AG and thus explained sedative effects of this compound in a cannabinoid receptor 1 and 2 double knockout mouse (Sigel et al, PNAS,2011). This positive modulatory action of 2-AG and other related endocannabinoids can only be seen at very low concentrations of GABA, thus excluding synaptic GABAA receptors operating at maximal GABA concentrations as a possible target. Accordingly in their PNAS paper the authors investigated the extrasynaptic delta-subunit containing GABAA receptor types in addition to the synaptic (a1b2g2) receptors. They found in their first paper that beta2-, but not beta1- containing receptors can be potentiated by 2-AG and that this potentiation can be reduced by half if two different beta subunits (concatenated) are present in the receptor a1-b2-a1-g2-b1. They also reported in PNAS amino acid residues (AARs) mediating the selectivity of 2-AG for the beta2 subunit (e.g. M294, L301, F439, which after mutation to the corresponding AAR in the beta 1 subunit significantly reduced potentiation whereas mutation V436T abolished it). In their next paper in J.Neurochem (2013) Baur et al mutated 30 AARs in the predicted binding site for 2-AG on the beta2 subunit to cystein and found that the transmembrane segment M4 and the intracellular portion near M3 and M4 contain critical amino acids, whose mutation reduces potentiation (% of control): V302C (43%), W428C (44%), S429C (49%),F432C(63%), V436T(4%), F439L(31%), V443C(49%). In the J.Neurochem paper Baur et al have shown that 1. NA-glycine and NA-serine are superior to 2-AG for the GABA-response potentiation (280%, 200% and 80% over control, respectively). 2. mutation b2V436T, which abolishes action of 2-AG also abolishes action of NA-glycine and 3. the action of 2-AG and NA-glycine is not additive (indicative for the same binding site). The present paper submitted to the PeerJ offers only 2 small additions to these previously published facts. First, it shows that some mutations do not affect peak potentiation by NA-glycine but reduce the steady-state phase. Second, it provides a hypothetical explanation for the different kinetics of 2-AG- and NA-glycine- potentiation through their different hydrophobicity and membrane solubilisation behaviour (measured as critical micelle concentration, CMC). I have two alternative suggestions how to improve value of the present manuscript. 1) re-write this story in a format of review where previous findings will be extensively described and few new additions and hypothesis (lipophilicity) will be added. 2) add more experiments with other endocannabinoids, which differ in their lipophilicity (e.g. NA-serine, NA-GABA, 1-AG, AEA, AA). As pH of endocannabinoids is critically important for the potentiation or inhibition of glycine receptors (Yevenes and Zeilhofer, PlosOne, 2011), this issue should be also adequately considered. Analysis of neuronal GABAA receptor modulation by 2-AG and NA-glycine can be added.

Major concerns.
1. Concentration-response relationship for the NA-glycine (Figure 2) contains recordings from 4 Xenopus oocytes. Data shown on this plot are probably containing the same values as in J.Neurochem paper 2013 (Baur et al) in Fig 3 (n=3). This is unacceptable (I would suggest simply to cite previous paper).
2. The course of the potentiation by 2-AG shown in J.Neurochem paper 2013 Fig.5 looks very similar (the legend is almost identical) to figure 9 of the present paper (this is shown now for the NA-glycine). According to my measurement, potentiation now represents 133% over control (NA-glycine) and not 280% (average) as shown in Figure 3 of their J.Neurochem paper.
3. Introduction. Lines 39-51 are identical (self-plagiarism) to the first paragraph of the J.Neurochem paper 2013.
4. line 60 Intro.”It has poor affinity for CB1 receptor …”. Add to it: “EC50>10µM, and to TRPV1 (capsaicin receptor), EC50>10µM. These potencies are not very far away from the modulatory potency at the GABAA receptor (ca 3µM for 2-AG and 1-10µM for NA-glycine) or glycine receptor (68µM).
5. line 64, add after “(Yevenes and Zeilhofer, 2011)”: potentiating alpha1 and inhibiting alpha2 and alpha3-containing glycine receptors.
6. Lipophilicity of 2-AG and NA-glycine should be measured and provided as 1-octanol/water partition coefficient.
7. Figure 8a shows representative trace of potentiated by NA-glycine GABA response. This response has very slow onset and reaches maximal amplitude value after 1min of application. This is contradictory to the averaged values from the same experiments, where peak amplitude (reached within 20s of application) is larger than the measurement taken at the end of application (1min point). This picture could be exchanged for a more typical one. Scale both control responses (left and right) to the same size, which will allow the reader to notice reduction of potentiation to 50% of control after mutation.
8. Figure 8b and lines 206-207 page10. Unless I have misunderstood something, AARs are given wrong here for the a1b2S428Cg2 (should be W), a1b2R429Cg2 (should be S) and a1b2Y443Cg2 (should be V). Authors state that these experiments showed that mutations do not affect action of NA-glycine in contrast to 2-AG, but inspecting their numbers in J.Neurochem paper, I came to the conclusion that reduction of potentiation (when steady-state phase of the response is considered) is very similar or even larger in case of NA-glycine for all mutations except for the AAR V443C. Significance levels (when compared to the WT) should be provided on the Fig.8b.
9. page 12, line255-258. Logic of this conclusion is unclear. If two modulators are taken at their maximal concentrations I would expect their common application yielding an additive response if they interact with different binding sites, and non-additive if they interact with the same site. The latter is true for the NA-glycine and 2AG.
10. Demonstration of no effect of endocannabinoids at beta1-containnig GABAAR lacks novelty (published in PNAS paper) and can be omitted here as well as experiments with concatenated receptors. Only results from beta2-containing receptors should be presented. Instead, it would be very important to test whether extrasynaptic GABAAR types, for example containing alpha4, alpha5 or alpha 6 subunit are also potentiated by the 2-AG and NA-glycine. A study on glycine receptors showed that the alpha subunit type determines the direction of modulation (Yevenes and Zeilhofer).
11. Fig.5 Block of 2-AG-modulation by the DEA (which does not produce potentiation of GABA-response per se) was already published previously (Figure 4 of J. Neurochem paper). Now the authors show that the NA-glycine action is also blocked by DEA but at much higher concentrations. It is not clear whether the steady state (end of GABA-application period) or the peak amplitude was taken for the construction of concentration-response curves here. What was the kinetic of the block? How long must DEA be applied in the middle of GABA-2AG or GABA-NA-glycine application to achieve the block? How do mutation b2S429C or b2F439L affect this block by DEA? These or other questions could be addressed to make the present study more interesting and novel. Simple repetition of previously published protocols or even presentation of the same data bores the informed reader.
12. In PNAS, Sigel et al., one can read, that modulation of GABA-currents by 2-AG was not detected in native neurons in a brain slice preparation. This issue should be again discussed or investigated (does NA-glycine affect gabaergic sIPSCs in slices or GABA-evoked currents in neurons isolated from slices?).
13. The authors conclude in the last paragraph of discussion: “ No matter what the exact mode of interaction of NA-glycine with the GABAA receptor is, this agent (NA-glycine) represents by far the more potent positive allosteric modulator than 2-AG, although the latter is more abundant in brain.” Indeed, mechanisms of NA-glycine action remained questionable for the reader and may be the authors are right calling them unimportant at the end of their manuscript. With the second part of this sentence I cannot agree. In PNAS Sigel and colleagues calculated the 2-AG modulatory potency (EC50 =2µM), in J.Neurochem EC50=2.9µM. In J.Neurochem and in the present study EC50 of NA-glycine was estimated to be between 1 and 10µM (page7, line143, page9, line187). Thus, potencies of these two agents from pharmacological point of view are roughly similar. Modulatory efficacy of NA-glycine (maximal potentiation of GABA-response) is indeed higher than the efficacy of 2-AG (280% vs 80%, respectively, Fig.3, J.Neurochem,2013 paper), but these experiments are not shown in the present study, therefore they cannot be discussed in the concluding remarks.
14. It is not clear why, given such variability of responses to NA-glycine (EC50 between 1 and 10µM), the authors did not attempt to investigate more precisely this variability (e.g. whether apparent desensitisation and open channel block are voltage-, temperature-, cAMP-or pH-dependent). Measurements of EC50 may contain artefacts due to the slow distribution of substances around big oocytes and interplay between concentration ramp and desensitisation. More precise evaluation of desensitisation can be done in native neurons or in HEK293 cells, measurement of EC50 with different techniques could be compared.
Minor concerns
1. Na-glycine should be corrected through the text and pictures to NA-Gly or NA-Glycine
2. Fig.6 panels a and b are mixed-up.
3. Legend Fig 7: word “receptors” 2 times in a row on line 2
4. Fig 9 legend: replace microM with µM
5. In the method section, please, indicate from which species GABAA receptor subunits were used for the expression in Xenopus Oocytes (human, rat or mouse)?
6. page4, line 64 in“Yvenes” letter “e” is missing

Experimental design

n/a

Validity of the findings

n/a

·

Basic reporting

line 54. "specifically on the two b2 subunit containing receptor pentamers". Not clear - "two b2 subunits in a pentameric receptor"?

line 222. "This again points to the differences of 2-AG and NA-glycine in the way they dissolve and segregate into the membranes." Please clarify 2 points. First, does "This" refer to the absence of an "off response" (i.e. a lack of fast block)? This could, of course, simply reflect a much higher unblocking rate that is missed in the relatively slow perfusion of oocytes. Second, how is this the result of membrane partitioning? It could be that channel block is simply absent in one case?

line 234. "may reflect the better water solubility of NA-glycine over 2-AG at low concentrations" and line 263 "differential solubilisation of NA-glycine and 2-AG with Xenopus oocytes may account for some of the effects observed in this study. The way these lipids are organized in an aqueous environment will affect entry of the molecules into the bilayer, binding equilibrium, and the way the receptor is occupied." Please clarify the reasoning behind the statements - is it suggested that self-association in the membrane will alter the activity of the different endocannabinoids in the effect compartment?

line 241. The interpretation that there is a preferential effect on activation of monoliganded receptors is interesting. Have the authors used their constructs with one GABA-binding site eliminated to examine this question directly?

Figure 2 legend. Describe the meaning of the bars above the traces. Describe the curve shown in part b.

Figure 3. Please show the abscissa labelled in terms of (approximate) fractional activation as well as [GABA] (or at least provide the mean Hill parameters for GABA activation in the legend so the reader can do it herself).

Figure 4. Define K and L (e.g. K = association rate constant/dissociation rate constant?).

Figure 6. Panels a & b reversed in legend?

Experimental design

Sound - no comments

Validity of the findings

The fact that there are two binding sites in the receptor suggests that the interpretation of the experiments in which both 2AG and NAG are added simultaneously could be complex. What are the properties of heteroliganded receptors? Is it possible to conclude that the results directly address the question of whether the two compounds bind to the same site?

The experiments in which DEA is used to inhibit responses to 2AG are in part confounded by the additional information on CMC for 2AG. At 15 microM the possibility exists that 2AG micelles are sequestering DEA and so the actual activity of the inhibitor is lower. Have the experiments been done using two relatively low concentrations of 2AG (as were done with NAG); this would be more likely to provide an interpretable answer.

As a related question, does the CMC indicate that the apparent EC50 for 2AG may be affected by the aqueous free concentration of 2AG?

Figure 8. The time course of the response is indeed interesting, and as the authors point out may be difficult to explain as a consequence of altered binding. Indeed, the results suggest that some of these residues may be involved in transduction of effects.

Additional comments

This MS contains some very interesting observations, continuing the line of research reported in two earlier papers from this group. The MS is generally well written and understandable. I have some comments about the experiments and interpretations, and a few comments on basic reporting.

Reviewer 3 ·

Basic reporting

A clear, well-written paper. The following are some relatively minor points.

Lines 58-60: Could the authors provide some context for these amounts of NA-glycine? That it is a small amount is inferred, but how do these levels compare to 2-AG and/or AEA?

Lines 60-61: What is known about the effects of NA-glycine on TRP channels?

Lines 136-138 & Fig. 2 legend: It is not stated explicitly by the authors in this early part of the Results section (or the corresponding figure legend) that what is being described is potentiation as a result of increasing NA-glycine concentration.

Line 191: Extra period at the end of the sentence.

Lines 200-202: Incorrect grammar. Suggest something like the following;
“Receptors containing two B2 subunits exhibited strong potentiation while receptors containing two B1 subunits showed very weak potentiation. Intermediate potentiation was observed in receptors containing one each, B1 and B2.”.

Line 217: When I first read this paragraph, I thought the authors were trying to distinguish between NA-glycine reaching the receptor by passive diffusion vs. a transport system (active transport system?). Eventually, I realized that they were just trying to demonstrate that the onset of the NA-glycine effect was slow; consistent with the transmitter having to go through the membrane to reach the receptor and were not trying to determine if this was via diffusion or transport. Perhaps the authors could revise this paragraph to make this point a little more clear.
Another issue with this section is that at the end of the paragraph, the authors state that this finding “points to the differences between of 2AG and NA-glycine…”, implying that the two transmitters differed in how long it took for them to exert their effects on the GABA receptor. In the Discussion however (lines 250-251), the authors state that “the onset of action for both drugs was found to be slow”. Can the authors please explain or correct this apparent discrepancy?

Line 240 & 251: Change “drugs” to “transmitters”. The term “drug” implies an exogenous agent and these are both endogenous neurotransmitters.

Lines 265-267: Can the authors provide a citation to support this statement?

Lines 269-271: This sentence does not make sense. Is the last part supposed to read “…the receptor site may be able to bind a surface flexible hydrophobic structures.”?

Figure 6 legend: The order of the (a) and (b) parts of the figure legend is the opposite of how the corresponding graphs appear in Figure 6.

Experimental design

No comments.

Validity of the findings

While there are some perplexing findings in terms of the DEA/2-AG interaction and the results from the co-application of 2-AG and NA-glycine, the experimental design is sound and the quality of the data appears to be excellent. Furthermore, the authors have considerable experience in this field. I am perfectly comfortable with presenting the data in its current state along with the speculation about the potential role of differences in solubility and lipid fluidics.

Couple of minor points.
Lines 176-179: The authors’ hypothesis that issues of water/lipid solubility may explain the unexpected results for the preceding experiments is perfectly valid. It appears that the DEA/NA-glycine results are consistent with a non-competitive interaction mechanism, but that the DEA/2-AG results cannot be readily explained. Is it possible that 2-AG could directly open the GABA receptor? The authors report that 3uM NA-glycine produces no direct current, but what about 15uM 2-AG?

Figure 2b: It appears that receptors with the β1 subunit actually underwent depression during NA-glycine treatment. Is this a correct interpretation of that figure? If so, the authors should address that potentially interesting finding.

Additional comments

A very good manuscript.

Reviewer 4 ·

Basic reporting

The manuscript fully meets the required standards of PeerJ concerning "Basic Reporting".

Experimental design

The manuscript fully meets the required standards of PeerJ concerning "Experimental Design".

Validity of the findings

The manuscript widely meets the required Standards of PeerJ concerning "Validity of the Findings". However, the authors should add one sentence in the introduction clearly stating the aim of the study, instead of merely paraphrasing it.

Additional comments

This is a well conducted study with clearly novel findings.

---

## Round 0.2 · accepted · Accept

Thank you for revising the above mentioned manuscript and answering all the reviewers questions.